# Contact-aware Human Motion Forecasting

**Wei Mao**[1], **Miaomiao Liu**[1], **Richard Hartley**[1], **Mathieu Salzmann**[2,3]

[1]Australian National University; [2]CVLab, EPFL; [3]ClearSpace, Switzerland

{wei.mao, miaomiao.liu, richard.hartley}@anu.edu.au, mathieu.salzmann@epfl.ch

## Abstract

In this paper, we tackle the task of scene-aware 3D human motion forecasting, which consists of predicting future human poses given a 3D scene and a past human motion. A key challenge of this task is to ensure consistency between the human and the scene, accounting for human-scene interactions. Previous attempts to do so model such interactions only implicitly, and thus tend to produce artifacts such as "ghost motion" because of the lack of explicit constraints between the local poses and the global motion. Here, by contrast, we propose to explicitly model the human-scene contacts. To this end, we introduce distance-based *contact maps* that capture the contact relationships between every joint and every 3D scene point at each time instant. We then develop a two-stage pipeline that first predicts the future contact maps from the past ones and the scene point cloud, and then forecasts the future human poses by conditioning them on the predicted contact maps. During training, we explicitly encourage consistency between the global motion and the local poses via a prior defined using the contact maps and future poses. Our approach outperforms the state-of-the-art human motion forecasting and human synthesis methods on both synthetic and real datasets. Our code is available at `https://github.com/wei-mao-2019/ContAwareMotionPred`.

## 1 Introduction

Human motion prediction has a broad application potential covering human robot interaction [17], autonomous driving [23], virtual/augmented reality (AR/VR) [28] and animation [31]. As such, it has been an active research topic for decades [4, 27, 30, 35]. Nevertheless, most methods [19, 1, 22, 32, 9, 18, 21, 5] disregard the fact that humans evolve in 3D environments, thus ignoring the human-scene interactions. By contrast, in this work, we tackle the task of scene-aware human motion forecasting, which aims to incorporate the scene context to predict future 3D human motions.

While some recent works in human motion prediction [6, 8] and human synthesis [33] have started to explore the use of scene context, they do so implicitly, by taking an embedding of a 2D scene image [6], a 3D scene [33], or a specific object [8] as an additional input to their model. While such embeddings encode valuable information, they do not provide precise cues to help placing the human in the scene. In the context of synthesizing a static human body in a scene, some efforts have nonetheless been made to incorporate more precise information, such as the distance between every scene point and the closest vertex on body surface [36], or a semantic scene label, e.g., floor, sofa, at every vertex on the 3D body mesh [12]. However, while such representations allow one to place a human in the scene, they remain under-constrained for motion prediction. Specifically, they impose neither temporal consistency, nor consistency between local pose changes and global motion. As such, they yield artifacts such as "ghost motion".

In this paper, we address this by explicitly modeling the contact between the human body joints and the scene. To this end, we introduce per-joint contact maps that encode the distance between each joint and every 3D scene point. Such contact maps constrain both the global motion and the local human pose, thus avoiding the "ghost motion" issue. As illustrated in Fig 1, our human motion

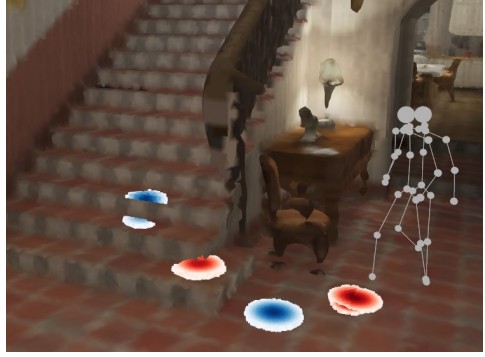 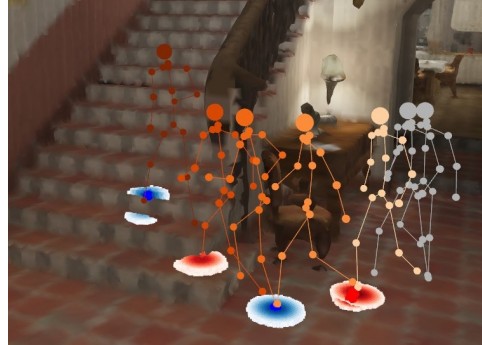

(a) predict the future contact maps      (b) forecast the future human poses

Figure 1: **Our approach.** Given the historical motion, depicted as gray skeletons, and the 3D scene, (a) our model first predicts per-joint contact maps in the future frames, shown with color maps. Here, for visualization purpose, we only show the contact maps of the left foot (blue) and right foot (red). (b) Conditioned on the predicted contact maps, we then forecast future poses, shown as orange skeletons. The per-joint contact maps provide strong cues about the 3D human joint locations.

prediction framework therefore consists of a two-stage pipeline that first predicts the future contact maps and then forecasts the future human poses by conditioning them on the predicted maps.

Specifically, we obtain a compact temporal representation of the historical contact maps using a similar Discrete Cosine Transform (DCT) encoding strategy to that of [22] for human motion, and use a feed-forward 3D scene encoding network, i.e., PVCNN [20], to predict the future contact maps from the scene point cloud, past human motion and the historical contact maps. To then forecast human motion, we retain the closest contact points for every human joint according to the predicted contact maps and forecast the global translations and then the local poses given these points and the past human motion. Consistency between the global translations and the local poses is achieved via a prior defined using the predicted human poses and the contact points.

Our contributions can be summarized as follows. (i) We introduce a distance-based per-joint contact map that captures fine-grained human-scene interactions to avoid generating unrealistic human motions. (ii) We further propose a two-stage pipeline whose first stage models the temporal dependencies of past contact maps and predicts the future ones, and whose second stage forecasts the future human motion conditioned on these contact maps. Our experiments on both synthetic and real datasets demonstrate the benefits of our approach over the state-of-the-art human motion forecasting and human motion synthesis methods.

## 2 Related Work

**Human motion prediction.** Modeling 3D human motion has been a long-standing research goal [4, 27, 30, 35]. While traditional methods [4, 35], relying on either Hidden Markov Models [4] or the Gaussian process latent variable model [35], can tackle periodic and simple non-periodic motions, such as walking and golf swing, more complex motions have been shown to be better modeled via deep learning frameworks [19, 1, 22, 32, 9, 18, 21, 5], which can be roughly categorized into feed-forward models [19, 22, 21] and recurrent networks [1, 32, 9, 18, 5] according to their temporal-spatial encoding strategies. Despite the success of these methods at forecasting complex motions, they typically only predict local poses, disregarding global motion and any scene information. In this paper, we seek to predict future human motions that are consistent with the 3D scenes they are performed in.

Recently, a few works [8, 6] have started to incorporate scene context in motion forecasting. In particular, Corona *et al.* [8] introduced a semantic-graph model that extracts a joint embedding of the human pose and an object of interest, such as a cup. This method, however, is ill-suited to model interactions with the whole scene itself, for example the floor or stairs that the person touches while walking. In [6], Cao *et al.* proposed a multi-stage pipeline that breaks down the motion forecasting into three sub-tasks: predicting a 2D goal, planning a 2D and 3D path, forecasting the 3D poses

following the path. To this end, they extracted a scene representation using 2D images and let their model learn the scene constraints implicitly. This strategy, however, cannot handle scene occlusions and does not enforce consistency between the local and global motion. More importantly, both these methods aim to learn interactions implicitly. By contrast, our contact maps explicitly encode the human-scene interactions, leading to direct constraints at the human joint level.

**Scene-aware human synthesis.** Synthesizing a realistic human in a 3D scene has recently gained an increasing popularity [36, 37, 12, 34, 33, 10]. A cornerstone in the success of these methods is the modeling of human-scene interactions. To achieve this, many of these works [37, 34, 33] follow a similar approach to [6], modeling the human-scene interactions implicitly, via either Generative Adversarial Networks (GANs) [34], or Variational Autoencoders (VAEs) [37, 33]. Three methods [36, 12, 10] nonetheless exploit explicit representations of human-scene interaction. In particular, POSA [12] uses a body-centric representation of the human-scene interaction where a semantic scene label, e.g., floor, sofa, is assigned to every human mesh vertex. This label encodes the contact probability to the scene surface and the corresponding semantic scene label. However, this semantic representation does not provide any information about the 3D location of the human body, and is thus ill-suited to the motion forecasting task. In PLACE [36], Zhang *et al.* introduce a contact representation based on Basic Point Sets (BSP) [26]. Specifically, given a set of basic 3D scene points, they represent the human-scene interaction using the minimum distance from every such point to the human body surface. As in the semantic-based case, this strategy only gives a weak prior on the human pose, as it does not explicitly defines which joint should be in contact with which scene point. In SAMP [10], their approach only models a coarse interaction of the human with a given object in the final frame by predicting the final root location and orientation. Such coarse interaction however cannot constrain the poses at intermediate frames. By contrast, our per-joint contact maps provide a more detailed contact information for every human joint at each future frame.

**Hand-object interaction** Although hand-object contact relationships have already been studied for the task of grasping [2, 29, 3, 14], existing methods cannot be naively applied to human-scene interactions because their object-centric contact relationships tend to be static across time. For example, when we are using a hammer, we will grasp the handle tightly, and thus the contact region between our palms and the hammer does not change across time. By contrast, our human-scene contact maps change across the frames for almost all human activities. This motivates us to propose a distance-based per-joint contact map at each frame.

## 3 Approach

Let us now introduce our approach to scene-aware 3D human motion forecasting. Following previous work [22], a sequence of $P$ past human poses is represented as $\mathbf{X} = [\mathbf{x}_1, \mathbf{x}_2, \cdots, \mathbf{x}_P] \in \mathbb{R}^{P \times J \times 3}$, where $\mathbf{x}_p \in \mathbb{R}^{J \times 3}$ encodes the 3D locations of all $J$ joints at time $p$ in a *global* reference frame. The 3D scene is represented as a set of 3D points $\mathbf{S} \in \mathbb{R}^{N \times 3}$. Given the historical motion $\mathbf{X}$ and the 3D scene $\mathbf{S}$, our goal is then to forecast $T$ future human poses $\mathbf{Y} = [\mathbf{x}_{P+1}, \mathbf{x}_{P+2}, \cdots, \mathbf{x}_{P+T}] \in \mathbb{R}^{T \times J \times 3}$. To this end, we introduce a two-stage pipeline that first predicts future human-scene contact maps, and then forecasts the future poses by conditioning them on these contact maps. An overview of our pipeline is shown in Fig. 2. Below, we present our contact representation, and our pipeline to predict future contact maps and future motion.

### 3.1 Per-joint Scene Contact Map

We represent human-scene contact using the distances between the human joints and the scene points. Specifically, given a human pose $\mathbf{x}$ and a 3D scene $\mathbf{S}$, we first compute the per-joint distance map $\mathbf{d} \in \mathbb{R}^{J \times N}$, where each entry $\mathbf{d}_{jn}$ encodes the $\ell_2$ distance between the $j$-th human joint and $n$-th scene point, i.e.,

$$\mathbf{d}_{jn} = \|\mathbf{x}_j - \mathbf{S}_n\|_2 \, , \tag{1}$$

where $j \in \{1, 2, \cdots, J\}$ and $n \in \{1, 2, \cdots, N\}$.

In the distance map, the scene points that are less relevant to a human joint, because they are far away from it, have a higher value. This will tend to give them more influence when used in a deep neural network and may in turn cause issues when training our contact prediction module. To address this, we normalize the distance map to obtain a continuous contact map $\mathbf{c} \in \mathbb{R}^{J \times N}$ whose elements are

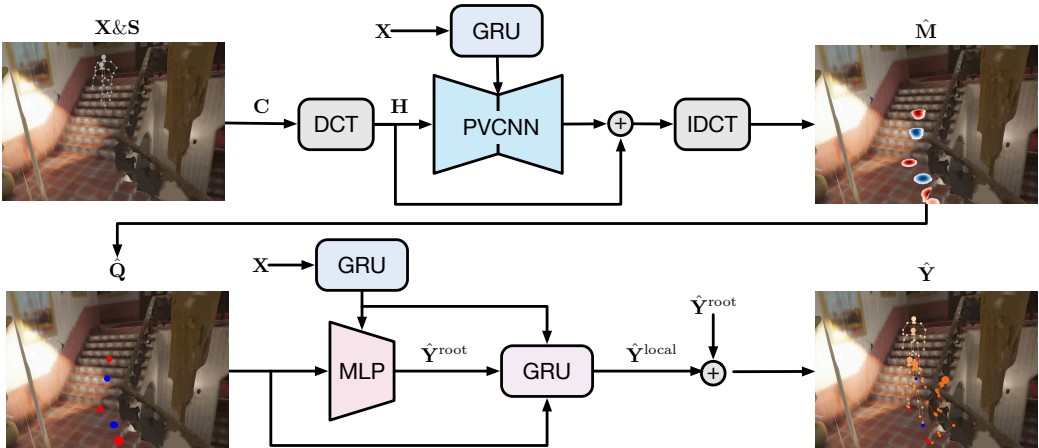

Figure 2: **Network architecture.** Our contact prediction network (top row) takes as input the scene points $\mathbf{S}$, the past contact maps $\mathbf{C}$ and the past human motion $\mathbf{X}$ to predict the future contact maps $\hat{\mathbf{M}}$. From these contact maps, we then extract the per-joint contact points $\hat{\mathbf{Q}}$ for future time instants. Conditioned on these contact points, our motion forecasting network (bottom row) forecasts the global translations $\hat{\mathbf{Y}}^{\text{root}}$ and then the local poses $\hat{\mathbf{Y}}^{\text{local}}$.

defined as

$$\mathbf{c}_{jn} = e^{-\frac{1}{2} \frac{\mathbf{d}_{jn}^2}{\sigma^2}} \, , \tag{2}$$

where the constant $\sigma$ is the normalizing factor. In such a contact map, closer scene points have higher values than far-away ones, whose values will be very close to zero.

Given the 3D scene $\mathbf{S}$ and past human poses $\mathbf{X}$, we compute the sequence of past contact maps $\mathbf{C} = [\mathbf{c}_1, \mathbf{c}_2, \cdots, \mathbf{c}_P] \in \mathbb{R}^{P \times J \times N}$ as described above. Our goal then is to predict the future contact maps $\mathbf{M} = [\mathbf{c}_{P+1}, \mathbf{c}_{P+2}, \cdots, \mathbf{c}_{P+T}] \in \mathbb{R}^{T \times J \times N}$. In the next section, we introduce a contact map prediction module to do so.

## 3.2 Contact Prediction Network

Since our contact maps are based on distances, they are smooth over time. Therefore, similar to [22] for human motion, we adopt a temporal encoding strategy based on the Discrete Cosine Transform (DCT). With the DCT, a sequence is represented as a linear combination of a set of pre-defined cosine bases. By discarding the high-frequency components, the DCT provides a compact representation, which nicely captures the smoothness of the sequence.

More formally, let us denote a $P$-frame sequence of contact values between the $j$-th human joint and the $n$-th scene point as $\tilde{\mathbf{c}}_{jn} = [\mathbf{c}_{1jn}, \mathbf{c}_{2jn}, \cdots \mathbf{c}_{Pjn}]$. This sequence can be fully represented using $P$ DCT coefficients, with the $l$-th one given by

$$\mathbf{h}_{ljn} = \sqrt{\frac{2}{P}} \sum_{p=1}^{P} \mathbf{c}_{pjn} \frac{1}{\sqrt{1 + \delta_{l1}}} \cos\left(\frac{\pi}{2P}(2p - 1)(l - 1)\right) \, , \tag{3}$$

where $l \in \{1, 2, \cdots P\}$ and $\delta_{ij}$ denotes the *Kronecker* delta function, i.e.,

$$\delta_{ij} = \begin{cases} 1 & \text{if } i = j \\ 0 & \text{if } i \neq j \, . \end{cases} \tag{4}$$

Given these DCT coefficients, the sequence in the original space can be obtained via the Inverse Discrete Cosine Transform (IDCT) as

$$\mathbf{c}_{pjn} = \sqrt{\frac{2}{P}} \sum_{l=1}^{L} \mathbf{h}_{ljn} \frac{1}{\sqrt{1 + \delta_{l1}}} \cos\left(\frac{\pi}{2P}(2p - 1)(l - 1)\right) \, , \tag{5}$$

where $L$ is the number of DCT coefficients used. For a lossless reconstruction, $L = P$, but when the sequence is smooth, one can ignore the coefficients of the high-frequency DCT bases and thus use $L < P$ at a negligible loss.

Recall that our goal is to predict the future $T$ contact maps $\mathbf{M} = [\mathbf{c}_{P+1}, \mathbf{c}_{P+2}, \cdots, \mathbf{c}_{P+T}]$ from the past $P$ ones $\mathbf{C} = [\mathbf{c}_1, \mathbf{c}_2, \cdots, \mathbf{c}_P]$. To leverage the DCT representation, we then reformulate this problem as learning a mapping from the DCT coefficients of the past contact maps to those of the future ones. Specifically, following the padding strategy of [22], we first repeat the last contact map $\mathbf{c}_P$ $T$ times to create a sequence of $P + T$ contact maps $\mathbf{C}' = [\mathbf{c}_1, \mathbf{c}_2, \cdots, \mathbf{c}_P, \mathbf{c}_P, \cdots, \mathbf{c}_P] \in \mathbb{R}^{(P+T) \times J \times N}$. We then compute the DCT coefficients of this sequence and define it as $\mathbf{H} \in \mathbb{R}^{L \times J \times N}$. Note that for each joint we only retain the first $L$ DCT coefficients. Our goal then is to learn a residual between these DCT coefficients $\mathbf{H}$ and those of the real sequence $[\mathbf{c}_1, \mathbf{c}_2, \cdots, \mathbf{c}_{P+T}]$.

We formulate this as a point-cloud encoding problem. That is, for each scene point, we regard the corresponding DCT coefficients of all joints as its feature. Thus, every scene point has an initial feature vector of size $LJ$. Our contact prediction network then takes as input the 3D scene points $\mathbf{S}$, the DCT features $\mathbf{H}$ and the past human poses $\mathbf{X}$, and outputs the future contact maps $\hat{\mathbf{C}} = [\hat{\mathbf{c}}_1, \hat{\mathbf{c}}_2, \cdots, \hat{\mathbf{c}}_{P+T}]$ as

$$\hat{\mathbf{C}} = \text{IDCT}(\mathbf{H} + \mathcal{F}(\mathbf{S}, \mathbf{H}, \mathcal{G}_x(\mathbf{X}))) , \tag{6}$$

where $\mathcal{F}$ represents the trainable point-cloud processing model and $\mathcal{G}_x$ is the GRU encoder.

Specifically, as shown in Fig. 2, we use the Point-Voxel CNN (PVCNN) [20] to encode the 3D scene with its DCT feature vectors. The PVCNN was designed to process a 3D point cloud. It incorporates voxel-based convolutions and point-based representations, leading to a memory- and computation-efficient structure for 3D data. We adapt the PVCNN to also take as input the past human poses $\mathbf{X}$, encoded by a Gated Recurrent Unit (GRU) [7], to predict a residual of the DCT coefficients. We then obtain the predicted contacted maps $\hat{\mathbf{C}} = [\hat{\mathbf{c}}_1, \hat{\mathbf{c}}_2, \cdots, \hat{\mathbf{c}}_{P+T}]$ via the IDCT.

Note that, here, we seek not only to predict the future contact maps but also to recover the past ones. To this end, we use the average $\ell_2$ loss between the ground-truth contact maps and the predicted ones. Formally, this loss is defined as

$$\ell_{\text{map}} = \frac{1}{(P+T)JN} \sum_{p=1}^{P+T} \sum_{j=1}^{J} \sum_{n=1}^{N} \|\mathbf{c}_{pjn} - \hat{\mathbf{c}}_{pjn}\|_2^2 . \tag{7}$$

## 3.3 Motion Forecasting Network

Given the future contact maps $\mathbf{M}$ and the past human motion $\mathbf{X}$, our human motion forecasting module aims to predict the future human poses $\mathbf{Y}$. To this end, we first extract the closest contact scene point to each joint from the contact maps. Given these contact points, we first use a simple neural network to predict the future path (global translations) and then forecast the future local poses with an RNN-based model. Let us discuss these steps in more detail.

**Contact points.** Since our focus now is to predict the 3D location of human joints, we propose to retain the most relevant scene point i.e., the closest one to each joint. Specifically, given the contact map at time step $p$ $\mathbf{c}_p \in \mathbb{R}^{J \times N}$ and the 3D scene points $\mathbf{S} \in \mathbb{R}^{N \times 3}$, we would like to find the scene point that is closest to each human joint. Let us denote the resulting contact points as $\mathbf{q}_p \in \mathbb{R}^{J \times 4}$, each row of which stores the 3D location of the scene point closest to the corresponding human joint together with a binary value indicating whether the joint truly is in contact with the scene or not. More formally, the contact point for joint $j$ is computed as

$$\mathbf{q}_{pj} = \begin{cases} [\mathbf{S}_k, 1] , & \text{where } k = \underset{n = \{1, 2, \cdots, N\}}{\text{argmax}} \mathbf{c}_{j,n} , & \text{if } \mathbf{c}_{j,k} > \epsilon \\ [0, 0, 0, 0] & \text{otherwise} , \end{cases}$$

where $\mathbf{S}_k$ is the 3D location of the $k$-th scene point, and $\epsilon$ is a threshold to determine whether the joint is in contact with the scene or not. We compute such contact points for the entire future sequence, which yields a sequence of contact points $\mathbf{Q} = [\mathbf{q}_{P+1}, \mathbf{q}_{P+2}, \cdots, \mathbf{q}_{P+T}] \in \mathbb{R}^{T \times J \times 4}$, where $\mathbf{q}_p \in \mathbb{R}^{J \times 4}$ is the contact points at the $p$-th time step.

**Motion forecasting.** As shown in Fig 2, our forecasting model first predicts the global translation in each frame and then the local motion given the global translations. Specifically, given the past motion $\mathbf{X}$ and the contact points $\mathbf{Q}$, we use a simple multilayer perceptron (MLP) to predict the future global translations $\mathbf{Y}^{\text{root}} \in \mathbb{R}^{T \times 3}$ as

$$\hat{\mathbf{Y}}^{\text{root}} = \mathcal{M}(\tilde{\mathcal{G}}_x(\mathbf{X}), \mathbf{Q}) , \tag{8}$$

where $\mathcal{M}$ represents the MLP and $\tilde{\mathcal{G}}_x$ is the GRU to encode the past motion.

The global translations are then fed into a GRU to predict the local pose at each future time step. Assuming that the past and future local pose sequences are represented by $\mathbf{X}^{\text{local}} = [\mathbf{x}_1^{\text{local}}, \mathbf{x}_2^{\text{local}}, \cdots, \mathbf{x}_P^{\text{local}}] \in \mathbb{R}^{P \times (J-1) \times 3}$ and $\mathbf{Y}^{\text{local}} = [\mathbf{x}_{P+1}^{\text{local}}, \mathbf{x}_{P+2}^{\text{local}}, \cdots, \mathbf{x}_{P+T}^{\text{local}}] \in \mathbb{R}^{T \times (J-1) \times 3}$, respectively, this can be expressed as

$$\hat{\mathbf{x}}_p^{\text{local}} = \hat{\mathbf{x}}_{p-1}^{\text{local}} + \mathcal{G}(\hat{\mathbf{x}}_{p-1}^{\text{local}}, \hat{\mathbf{x}}_p^{\text{root}}, \mathbf{q}_p, \tilde{\mathcal{G}}_x(\mathbf{X})) , \tag{9}$$

where $\mathcal{G}$ denotes the GRU to predict the furture poses, $p \in \{P+1, P+2, \cdots, P+T\}$, and $\hat{\mathbf{x}}_p^{\text{local}} \in \mathbb{R}^{(J-1) \times 3}$, $\hat{\mathbf{x}}_p^{\text{root}} \in \mathbb{R}^3$ and $\mathbf{q}_p \in \mathbb{R}^{J \times 4}$ are the local human pose, global translation and contact points at time $p$, respectively.

The global translation and local motion prediction modules are trained jointly. To this end, we use 3 loss terms. The first one is a global translation loss defined as

$$\ell_{\text{root}} = \frac{1}{T} \sum_{p=P+1}^{P+T} \|\mathbf{x}_p^{\text{root}} - \hat{\mathbf{x}}_p^{\text{root}}\|_2^2 , \tag{10}$$

where $\mathbf{x}_p^{\text{root}} \in \mathbb{R}^3$ and $\hat{\mathbf{x}}_p^{\text{root}} \in \mathbb{R}^3$ are the ground-truth and predicted global translations at time $p$.

The second loss accounts for the local human pose prediction and is expressed as

$$\ell_{\text{local}} = \frac{1}{T(J-1)} \sum_{p=P+1}^{P+T} \sum_{j=1}^{J-1} \|\mathbf{x}_{pj}^{\text{local}} - \hat{\mathbf{x}}_{pj}^{\text{local}}\|_2^2 , \tag{11}$$

where $\mathbf{x}_{pj}^{\text{local}} \in \mathbb{R}^3$ and $\hat{\mathbf{x}}_{pj}^{\text{local}} \in \mathbb{R}^3$ are the ground-truth and predicted local positions of the $j$-th joint at time $p$.

Finally, our third loss term encodes a contact prior based on the contact points. We define it as

$$\ell_{\text{contact}} = \frac{1}{TJ} \sum_{p=P+1}^{P+T} \sum_{j=1}^{J} \mathbf{q}_{pj4} \|\hat{\mathbf{x}}_{pj} - \mathbf{q}_{pj([1:3])}\|_2^2 , \tag{12}$$

where $\hat{\mathbf{x}}_{pj} \in \mathbf{R}^3$ is the predicted location of joint $j$ at time $p$, obtained by adding the predicted global translation at time $p$ to the corresponding local pose. $\mathbf{q}_{pj[1:3]} \in \mathbb{R}^3$ and $\mathbf{q}_{pj4} \in \{0, 1\}$ are the 3D coordinates of the contact scene point and the indicator value, respectively.

The overall loss is then expressed as

$$\ell_{\text{motion}} = \lambda_1 \ell_{\text{root}} + \lambda_2 \ell_{\text{local}} + \lambda_3 \ell_{\text{contact}} . \tag{13}$$

We use a stage-wise training strategy where the contact prediction network and motion forecasting network are trained separately. During training, the motion forecasting network is given the ground-truth contact points as input. At test time, we first compute the contact points from the predicted contact maps and then use these contact points to forecast the future human motion.

## 4    Experiments

### 4.1   Datasets

We evaluate our method on two datasets, GTA-IM [6] and PROX [11].

**GTA-IM.** The GTA Indoor Motion dataset [6] is a large-scale synthetic dataset that captures human-scene interactions. It consists of 50 different characters performing various activities in 7 different scenes. Each scene is a building, and each building has several rooms on one or more floors. The

| method | Path Error (mm) | | | | | Pose Error (mm) | | | | |
|---|---|---|---|---|---|---|---|---|---|---|
| | 0.5s | 1.0s | 1.5s | 2.0s | mean | 0.5s | 1.0s | 1.5s | 2.0s | mean |
| LTD [22] | 67.0 | 119.3 | 207.6 | 375.6 | 147.4 | 67.5 | 93.8 | 98.9 | 103.5 | 80.5 |
| DMGNN [18] | 82.7 | 158.0 | 227.8 | 286.9 | 156.2 | **47.5** | 69.1 | 85.6 | 95.3 | 64.9 |
| SLT* [33] | **45.9** | 117.0 | 186.7 | 267.1 | 121.8 | 70.8 | 181.4 | 150.2 | 196.0 | 112.6 |
| Ours | 58.0 | **103.2** | **154.9** | **221.7** | **108.4** | 50.8 | **67.5** | **75.5** | **86.9** | **61.4** |
| Ours w/o contact | 61.1 | 111.7 | 171.0 | 249.0 | 118.8 | 57.8 | 74.8 | 82.4 | 98.1 | 68.2 |
| Ours w/ GT contact | 52.4 | 77.8 | 95.8 | 129.5 | 74.1 | 49.8 | 64.8 | 70.4 | 78.3 | 58.2 |

Table 1: **Quantitative results on GTA-IM [6].** We report the MPJPE in millimeter for both the global translations (path error) and the local poses (pose error). The "mean" error was obtained by averaging over the 60 future time steps. The * indicates that we adapted the official SLT [33] code, designed for motion synthesis, to our task. We also show ablation results of our model without contact maps ("Ours w/o contact") and with ground-truth contact maps ("Ours w/ GT contact").

| method | Path Error (mm) | | | | | Pose Error (mm) | | | | |
|---|---|---|---|---|---|---|---|---|---|---|
| | 0.5s | 1.0s | 1.5s | 2.0s | mean | 0.5s | 1.0s | 1.5s | 2.0s | mean |
| LTD [22] | 117.8 | 232.0 | 346.9 | 461.2 | 236.3 | 156.0 | 273.9 | 387.7 | 497.9 | 273.3 |
| DMGNN [18] | 119.1 | 242.7 | 360.2 | 462.4 | 243.7 | 91.0 | 141.3 | 171.8 | 187.8 | 129.1 |
| SLT* [33] | 105.8 | 227.2 | 384.1 | 453.5 | 255.0 | 112.1 | 230.6 | 233.7 | 269.6 | 175.5 |
| Ours | **93.3** | **187.2** | **284.4** | **381.2** | **192.2** | **89.9** | **127.5** | **149.3** | **167.5** | **116.8** |
| Ours w/o contact | 104.9 | 196.5 | 290.0 | 385.5 | 200.1 | 90.3 | 135.4 | 160.5 | 184.1 | 122.4 |
| Ours w/ GT contact | 73.9 | 106.7 | 104.6 | 117.4 | 88.0 | 83.7 | 112.9 | 125.2 | 132.9 | 101.1 |

Table 2: **Quantitative results on PROX [11].** Our model outperforms baseline models by a large margin across all time steps.

dataset contains around 1 million RGB-D frames together with the corresponding 3D human poses. We use 4 of the scenes as our training set ("r001","r002","r003","r006") and the remaining 3 as our test set ("r010","r011","r013")[1]. To obtain the 3D point clouds of the different scenes, we register their depth maps from different videos sequences with the ground-truth camera extrinsic matrices. Following [6], we use 21 out of the 98 human joints provided by the dataset. The videos run at 30Hz. We train our models to observe the past 30 time steps (1 second) and predict the future 60 time steps (2 seconds).

**PROX.** Proximal Relationships with Object eXclusion (PROX) [11] is a real dataset captured using a Kinect-One sensor. It comprises 12 different scenes with 20 subjects interacting with the scenes. The dataset also provides SMPL-X parameters [25] as the ground-truth human pose and shape in each frame. Since these parameters were obtained by a frame-wise fitting algorithm, the motion sequences are jittery and thus ill-suited to our task. We therefore refine the dataset via a simple temporal optimization process to generate smooth motions. More details are provided in the supplementary material. Following [33], we use 8 scenes for training ("N3Library", "MPH112", "MPH11", "MPH8", "BasementSittingBooth", "N0Sofa", "N3Office", "Werkraum") and 4 scenes for testing ("MPH16", "MPH1Library", "N0SittingBooth", "N3OpenArea"). We use the 22 body joints of SMPL-X model. As for GTA-IM, the frame-rate of this dataset is 30 Hz, and we train our models to take the past 30 time steps as input and predict the future 60 steps.

## 4.2 Metrics, Baselines & Implementation

**Metrics.** We use the Mean Per Joint Position Error (MPJPE) [13] to evaluate both the global translations (path error) and the local motion (pose error).

**Baselines.** We compare our method with two human motion prediction models (LTD [22] and DMGNN [18]) and one scene-aware human motion synthesis method (SLT [33]). LTD [22] is a representative feed-forward method based on Graph Convolutional Networks (GCNs) [16]. DMGNN [18]

---

[1]Note that the dataset does not provide an official training-testing split. We use this split to balance the number of motion sequences in training and testing.

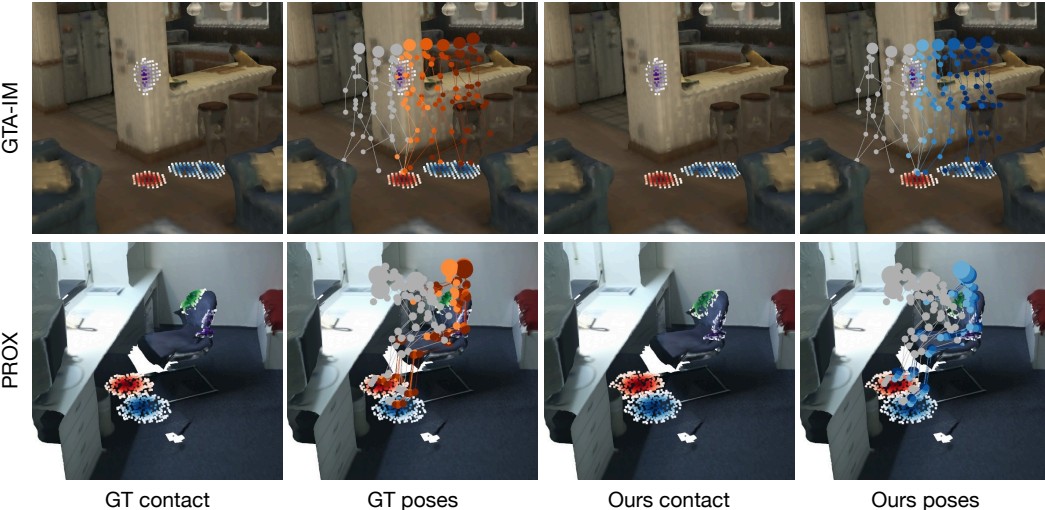

Figure 3: **Predicted contact maps on GTA-IM [6] (top) and PROX [11] (bottom).** We show the contact maps of four joints: left foot (blue points), left elbow (purple points), right foot (red points) and right elbow (green points). To show the influence of our contact maps, we also show the past motion (grey skeleton), ground-truth future motion (orange) and our predictions (blue).

is a state-of-the-art RNN-based approach for human motion prediction. SLT [33] is a stage-wise approach to synthesize long-term human motions. We used the official implementations of LTD [22] and DMGNN [18] to train them on our datasets. For SLT [33], we adapted the official code to our task e.g., we modified their model so as to take the past motion as input. The detail of these changes are provided in the supplementary material.

**Implementation details.** Our models are implemented in Pytorch [24] and trained using the ADAM [15] optimizer. Both our contact prediction network and motion forecasting one are trained for 50 epochs with learning rates of 0.0005 and 0.001, respectively. The training of each network takes about 12 hours on a 24GB NVIDIA RTX3090Ti GPU and the evaluation of one sample takes around 90 ms during testing. For both datasets, the normalizing factor $\sigma$, the number of DCT coefficients $L$ and the contact threshold $\epsilon$ are set to 0.2, 20 and 0.32, respectively. For the motion forecasting network, the loss weights $(\lambda_1, \lambda_2, \lambda_3)$ are set to $(1.0, 1.0, 0.1)$ for both datasets. For each motion sequence, we randomly sample 5000 scene points that are within 2.5 meters away from the root joint of the last observed pose. Additional implementation details are given in the supplementary material.

### 4.3 Results

**Quantitative results.** We provide quantitative results on GTA-IM and PROX in Table 1 and 2, respectively. Our approach outperforms the baselines for 3D paths and poses on both datasets across almost all time steps by a large margin. Specifically, the baseline models either perform well for the 3D path but comparatively poorly for the local poses, e.g., SLT [33] with an average path error of 121.8mm but the highest pose error on GTA-IM, or the reverse, e.g., DMGNN [18]. By contrast, our models produce more accurate 3D paths and poses than those of the baselines.

As an ablation study, we trained our human motion forecasting network without contact maps ("Ours w/o contact") and observe an increase of up-to 10mm in the mean path error and 7mm in the mean pose error. By contrast, using the ground-truth contact maps ("Ours w/ GT contact") yields a further performance boost, especially on the mean path error with a decrease of up-to 104mm. This indicates the effectiveness of conditioning the motion predictions on per-joint contact maps.

**Qualitative results.** We show our contact maps of four joints (left foot, left elbow, right foot and right elbow) in Fig. 3. Our model predicts accurate contact maps for diverse motions, such as "walking" (top) and "sitting down" (bottom). Note that, for the sample from GTA-IM [6] where the subject is about to walk around a corner, our contact maps precisely capture the contacts between the left elbow and the wall (shown as purple points on the wall), leading to accurate human motion predictions.

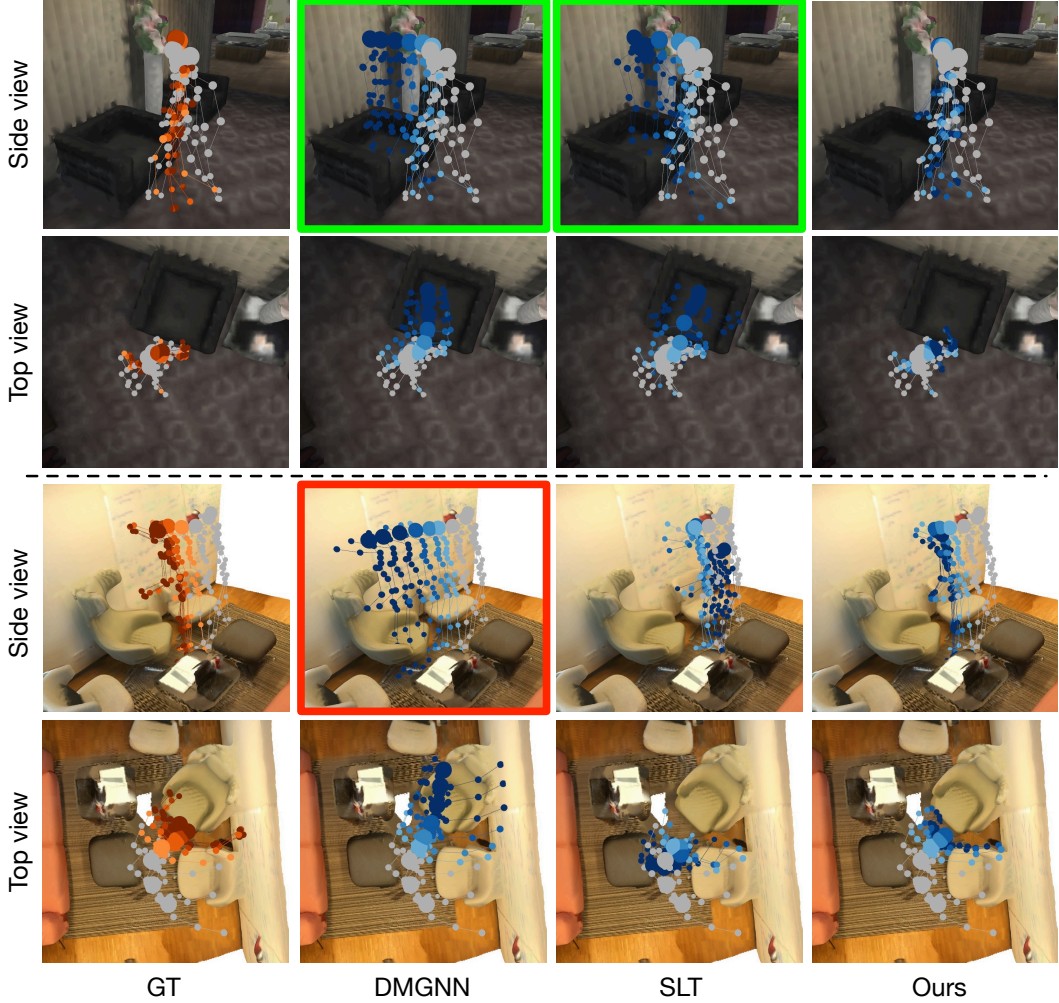

Figure 4: **Qualitative comparison.** We show the results of DMGNN [18], SLT [33] and our model on GTA-IM [6] (top) and PROX [11] (bottom). The past, ground-truth future and predicted future motions are shown in grey, orange and blue, respectively, and the shade of each color indicates the time step. The baseline methods, which either do not consider scene context [18] or implicitly model the human-scene interactions [33], tend to produce unrealistic human movements, e.g, ghost motion (highlighted with a red box) or walking through a sofa (highlighted with green boxes).

We further compare our results with those of the baselines in Fig. 4. We restrict this comparison to DMGNN [18] and SLT [33], which are quantitatively more accurate than LTD [22]. The complete comparison is included in the supplementary material. Due to the lack of explicit constraints on global motion and local movements, the baseline methods tend to produce unrealistic human motions, such as motions with almost no local movements but large global translations, i.e., ghost motion (highlighted with a red box), walking through a sofa (highlighted with green boxes). Thanks to our per-joint contact maps, our results are more plausible and closer to the ground truth.

## 5 Conclusion

In this paper, we have introduced a framework for scene-aware human motion forecasting that encourages consistency between global motion and local poses by exploiting human-scene contacts. To this end, we have proposed a per-joint contact map representation that captures the contact relationships between every human joint and the scene points. Our model consists of two stages. We first predict the per-joint contact maps given the motion history, and then forecast the future global

translations and local poses given the estimated future contact maps. Thanks to the explicit constraints provided by our per-joint contact maps, our method yields more plausible and more accurate future human motions than the state-of-the-art motion prediction or scene-aware human synthesis strategies.

**Limitations & Societal Impacts** One limitation of our work is that the quality of our predicted motions depends on that of the contact maps. As evidenced by our results with ground-truth contact maps, improving the contact map predictions translates in better motion predictions. This will therefore be one of our future research directions. Additionally, in many applications where the human shape matters, our joint-based contact map may not be enough to regularize the human surface. In our future work, we would like to extend our contact map to human surface. Furthermore, in real application, a potential risk of our method is that it may predict future motions that do not obey real physical rules, e.g., motion with imbalanced forces. For example, in the scenario of human-robot interaction where the agent i.e., a robot, needs to plan its actions according to the future human motion, such physically unrealistic future motion may lead to unsafe situations such as collision.

### Acknowledgements

This research was supported in part by the Australia Research Council DECRA Fellowship (DE180100628) and ARC Discovery Grant (DP200102274). The authors would like to thank NVIDIA for the donated GPU (Titan V).

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
