# OpenReview forum: "Contact-aware Human Motion Forecasting"
_NeurIPS.cc/2022/Conference — NeurIPS 2022 Accept_

### Official Review · Reviewer_cHVU · 2022-07-10

**Rating:** 7
**Confidence:** 3
**Soundness:** 3 good
**Presentation:** 3 good
**Contribution:** 3 good

**Summary:**

This paper promotes explicit contacts modeling when handling the challenging scene-aware human motion forecasting problem,

To achieve that, a dense scene-joint distance maps are utilized to densely model human dynamics when interacting with the static scenes,
followed by a novel discrete pre-processing with DCT to get sparse principal features, while also enabling residual motion prediction of
the contacts point in frequency domain.

A two-stage pipeline is assembled together to get better future motion predictions for all 3D body joints, with stage 1 combining GRU-based dynamics modeling and PVCNN-based 3D scene encoding for sequential contact distance maps prediction, and stage 2 carrying out contacts-guided sequential motion forecasting in stage 2.


**Questions:**

1. Related to Weakness 2, is this single closest contact priors
would be enough to guide the motion forecasting network? The single contact point looks like a special form of
signed distance function to me.

2. The motion forecasting network is still a very implicit design when leveraging contact points,
like for the global translation, the author seems to want the MLP module to learn root motion implicitly from all the closest points,
very similar to a learnable center point. If this is possible, I am wondering how this implicit design would be better than
a simple xyz averaging over all closest points belonging to the feet-nearby joints (maybe plus a certain offset).
Plus, it is not clear that in Figure 2, how does GRU take all three inputs to update the local joints offsets prediction?

3. How does number of scene points affect the prediction performance? Do we need to select all the scene points in order to train the stage 1 model? The author does not mention the computation efficiency clearly in the implementation section.

**Limitations:**

See above
Minor: Line 179, should be '...are then fed into...'

**Strengths And Weaknesses:**

**Strengths**
[Novelty]
As mentioned above, dynamic contacts modeling and the way to use it are the shining points in this paper, in which the authors leverage an effective point-joint distance field followed by a novel frequency transformation(DCT) to better capture sparse smooth motion patterns. Per-joint closest scene point is used to better guide the motion prediction. This design

[Completeness]
Three baselines and their proposed methods are validated in two common datasets, including both real and synthetic ones. They also provide both quantitative and qualitative results, including a video demo in the supp.

[Effectiveness]
The author gets consistently better motion forecasting results(global and local) on long-term motion predictions on two benchmarks compared to all the baselines.

**Weaknesses**
1. Firstly, even though the whole method seems to be novel, I do not think that line 49-54 is carefully written to
well capture the whole work, in (ii) the two-stage pipeline itself should not be considered as a key contribution,
it has overlaps with (i), also I did not see any unique technical contribution statements when tackling this conditional
motion synthesis task. The author needs a better clarification on the contributions.

2. Even though so many efforts(DCT/IDCT transformation, GRU models, PVCNN) have been conducted to get better
contact maps forecasting, it seems that only one nearest scene point is selected and used per joint point, I am
wondering whether such a heavy pipeline is really necessary.

3. The result in table 1 seems to show that the proposed method does not perform well enough on short-term predictions, even though it performs consistently better on predictions >= 1s

---

> ### Author Response · Authors · 2022-08-02
> **Responses to Reviewer cHVU**
>
> 1. Revising contributions
>
> __Response:__ Thank you for pointing this out. As shown in the revised manuscript, we would like to update our contributions as follow. i) We introduce a distance-based per-joint contact map that captures fine-grained human-scene interactions to avoid generating unrealistic human motions. ii) We further propose a two-stage pipeline whose first stage models the temporal dependencies of past contact maps and predicts the future ones, and whose second stage forecasts the future human motion conditioned on these contact maps.
>
> 2. Is it necessary to use GRU and PVCNN?
>
> __Response:__ The human-scene contact maps at different frames have strong temporal dependencies. For example, the left and right foot of a walking person will alternately touch the floor. The design of our contact map prediction network is necessary to capture such temporal dependencies. More specifically, we propose to represent human-scene contact based on the pair-wise distance between human joints and scene points. Such a representation is smooth across time and, as shown in [22], can be efficiently encoded via DCT. Although given a scene point cloud and a historical human motion, one could use a simpler classification network to directly distinguish whether a scene point is in contact with a certain joint or not at a particular future time step, it would be extremely hard for such a network to also model the temporal dependencies between the contact maps at different time steps.
>
> 3. Analysis about results at 0.5 second.
>
> __Response:__ As shown in Table 2, although our method does not perform well on the synthetic dataset GTA-IM for prediction at 0.5 second, it yields the best performance on the more complicated real world PROX dataset. We acknowledge that the benefits of our contact map could be limited on such short-term predictions because human motion in a very short future (e.g., less than 0.5 second) is highly constrained by the historical movements, e.g, the joint velocities and accelerations.
>
> 4. Is contact priors enough? (question)
>
> __Response:__ Contact points alone only constrain the position of the corresponding contact joints. However, when combined with other natural constraints, such as the length of the human limbs and the temporal smoothness of human motion, it can provide a guidance for the other joints as well. For example, when we only observe contact points for a joint in the first and third frames, the position of that joint in the second frame can only be somewhere in between. Note that these natural constraints are encoded in the global translation loss (Eq. 11) and the local pose error (Eq. 12).
>
> 5. Is global motion prediction necessary in motion forecasting network? (question)
>
> __Responses:__ Only a simple xyz averaging over all closest points of the joints near the feet joints may lead to ambiguities, because it is common to have root motion while the contact points remain the same. For example, during squatting, the contact points between our feet and the ground do not change while the location of our root joint (often defined at the pelvis) changes.
>
> In the second stage of our pipeline (Figure 2 bottom), we concatenate all three inputs as a long vector and then feed it to the GRU. As also expressed in Eq 10, we concatenate the latent feature of the historical motion $\mathbf{H}_x\in \mathbb{R}^D$ (from another GRU), the root joint location $\hat{\mathbf{x}}^{\text{root}}_p\in \mathbb{R}^3$ at frame $p$, the local human pose $\hat{\mathbf{x}}\_{p-1}^{\text{local}}\in\mathbb{R}^{(J-1)\times 3}$ at frame $p-1$ and the contact points $\mathbf{q}_p\in\mathbb{R}^{J\times 4}$ at frame $p$. Note that all variables are first flattened and then concatenated. The resulting long vector is then used to predict the local human pose at frame $p$.
>
> 6. Number of scene points v.s. performance (question)
>
> __Response:__ Since the maximum range of human motion in 2 seconds is fixed, it is not necessary to select all the scene points, which typically represent an entire building or room. During training, for each motion sequence, we randomly sample 5000 scene points that are within 2.5 meters away from the root joint of the last observed pose. Note that we only sample 5000 points because of GPU memory limitation. During testing, we can use a different number of scene points. In the table below, we compare the results of using different number of scene points given the pretrained model on GTA-IM. Using more scene points only yields a slight improvement in path error.
>
> | No. of Scene points | Mean path error (mm) | Mean pose error (mm) |
> |:---:|:---:|:---:|
> | 5000 | 108.2 | 61.4 |
> | 10000 | 106.1 | 61.2 |
> | 15000 | 106.2 | 61.1 |
> | 20000 | 106.2 | 61.1 |
>
> As to the computational efficiency, the evaluation of one sample takes around 90 ms during testing. Please see the revised manuscript for the updates.
>
> Thank you for pointing out the typo, please see the revised manuscript for the updates.

---

> > ### Comment · Reviewer_cHVU · 2022-08-08
> > **Followup feedbacks on rebuttal**
> >
> > Hi,
> >
> > Thanks for the clear explanation, it has addressed my questions.

---

### Official Review · Reviewer_QN5C · 2022-07-12

**Rating:** 5
**Confidence:** 4
**Soundness:** 3 good
**Presentation:** 3 good
**Contribution:** 3 good

**Summary:**

This paper proposes to tackle scene-aware 3D human motion forecasting by explicitly modeling the human-scene interactions, i.e., representing the contact between human body joints and scene points with a distance-based contact map. They also introduce a two-stage pipeline that first predicts the future contact map with the given motion history; then forecasts the future global translation and local poses. The proposed method can predict more physically plausible motions and avoid artifacts such as “ghost motion”.

**Main Contributions:** This paper proposes a contact map representation explicit modeling human-scene interactions and propose a two-stage framework to forecast human motions with given motion histories and 3D scenes.

**Questions:**

1. How do you use the motion features extracted by GRU in PVCNN? The PVCNN processes the 3D point cloud at point level, but the motion feature is the feature of the given motion history. Additionally, the notation $\mathbf{X}$ in Eq.6 is inaccurate, as $\mathcal{F}$ takes as input the latent feature of motion history instead of past human poses $\mathbf{X}$. Similar problems in Eq.9 and Eq.10.

2. The partial problem setting follows GTA-IM, which forecasts future human motions with a multi-stage pipeline. The quantitative evaluation should include the comparison between the proposed method and the variant of the method in GTA-IM.

3. How about the model's ability of forecasting long-term motions (more than 5 seconds)? The current setting only predicts the future motions in 2 seconds which is too short for humans to have distinguishable movements.

**Limitations:**

When considering the contact between human and scene, I think the shape-based human body representation, e.g., SMPL and marked-based representation, is more reasonable to model the contact between body surface and scene. This representation can produce a more fine-grained contact map, which thus can model more realistic details of human-scene interactions.

**Strengths And Weaknesses:**

- Strengths:

1.	This paper explicitly models the human-scene interactions with a contact map, which measures the distance between the human joints and the scene points. The contact map enables more physically plausible and realistic human-scene interaction generation.
2.	The proposed two-stage prediction pipeline disentangles contact prediction and human pose forecasting, thus capable of explicitly encouraging consistency between human motions and contact points in given 3D scenes.

- Weaknesses:

1. The contact map computed between joints and scene points is too coarse. It is more plausible to compute the contact map between body surface vertices and scene points because the human body surface rather than joints contact with the environment.
2. The contact map has been widely used in grasp generation tasks[ Jiang et al., ICCV2021; Brahmbhatt et al., CVPR2019 ]. I think you should discuss the contact map in the literature review. And the idea of using the contact map to model human-scene interactions is not very appealing.

---

> ### Author Response · Authors · 2022-08-02
> **Responses to Reviewer QN5C (Part 1. Weaknesses)**
>
> 1. Body surface contact
>
> __Response:__ In applications where body shape matters, we agree that it can be helpful to also consider the contact between the body surface and the scene. However, our per-joint contact map can also regularize the surface vertices given that the position of the human joint is often defined as a weighted sum of surface vertices (e.g., in SMPL-X [25]). Moreover, our joint-based contact map can also be easily extended to the surface contact map. Note that, at the time of submission, human-scene interaction datasets either only provide human skeletons, e.g., GTA-IM, or have noisy human body surfaces, e.g., PROX. The new dataset in [R1], which has recently been released, provides accurate scene and human body contacts. We would like to extend our method to this dataset in the future.
>
> 2. Discussion about grasping
>
> __Response:__ Thank you for the suggestion. Below we discuss the major difference between our human-scene contact map and the hand-object contact map in grasping. As shown in the revised manuscript, we will also include this discussion in the final version. Although hand-object contact relationships have already been studied for the task of grasping [R2,R3], existing methods cannot be naively applied to human-scene interactions because their object-centric contact relationships tend to be static across time. For example, when we are using a hammer, we will grasp the handle tightly, and thus the contact region between our palms and the hammer does not change across time. By contrast, our human-scene contact maps changes across the frames for almost all human activities. To capture such temporal dependencies, we propose to represent the human-scene contact based on the pair-wise distances between the human joints and the scene points, and use a DCT-based temporal encoding strategy to capture the cross-frame dependencies of contact maps. (Note that we assume that Jiang et al., ICCV2021 and Brahmbhatt et al., CVPR2019 refer to [R2] and [R3], respectively.
>
> [R1] Shimada, Soshi, et al. "HULC: 3D Human Motion Capture with Pose Manifold Sampling and Dense Contact Guidance." ECCV 2022.
>
> [R2] Jiang, Hanwen, et al. "Hand-object contact consistency reasoning for human grasps generation." ICCV. 2021.
>
> [R3] Brahmbhatt, Samarth, et al. "Contactdb: Analyzing and predicting grasp contact via thermal imaging." CVPR. 2019.

---

> > ### Author Response · Authors · 2022-08-02
> > **Responses to Reviewer QN5C (Part 2. Questions & Limitation)**
> >
> > 3. Details about GRU and PVCNN (question)
> >
> > __Response:__ We replicate the latent feature of motion history from GRU several times and then concatenate the resulting feature with the output of the PVCNN encoder. Specifically, given the output of the PVCNN encoder $\mathbf{H}_{\text{pcd}}\in \mathbb{R}^{N\times F}$, where $N$ is the number of points and $F$ is their feature dimension (point feature), and the latent feature of the historical motion $\mathbf{H}_x\in \mathbb{R}^D$ (motion feature), we first replicate the motion feature $N$ times and then concatenate the resulting feature with the point feature to form a feature matrix $\tilde{\mathbf{H}}\in \mathbb{R}^{N\times (F+D)}$. The PVCNN decoder takes $\tilde{\mathbf{H}}$ as input to produce a residual of the contact maps' DCT feature.
> >
> > Thank you for pointing out the inaccurate equations. As shown in the revised manuscript, we will also update these equations as follows in the final version.
> > $$\hat{\mathbf{C}} = \text{IDCT}(\mathbf{H} + \mathcal{F}(\mathbf{S},\mathbf{H},\mathcal{G}_x(\mathbf{X})))$$
> > $$\hat{\mathbf{Y}}^{\text{root}} = \mathcal{M}(\tilde{\mathcal{G}}_x(\mathbf{X}),\mathbf{Q})$$
> > $$\hat{\mathbf{x}}_p^{\text{local}}=\hat{\mathbf{x}}\_{p-1}^{\text{local}}+\mathcal{G}(\hat{\mathbf{x}}\_{p-1}^{\text{local}},\hat{\mathbf{x}}_p^{\text{root}},\mathbf{q}_p,\tilde{\mathcal{G}}_x(\mathbf{X}))$$
> > Here, $\mathcal{G}_x$, $\tilde{\mathcal{G}}_x$ refer to the GRU motion encoders.
> >
> > 4. Comparison to the variant of the method in GTA-IM. (question)
> >
> > __Response:__ To the best of our knowledge, the only work that uses the GTA-IM dataset and released their official code is [R4]. We compare their results with ours in the table below. Note that the original model of [R4] observes past 2D human poses to predict future motion. We adapted their code to take 3D past human motion as input.
> >
> > |  |  |  | Path |  |  |  |  |  | Pose |  |  |
> > |:---:|:---:|:---:|:---:|:---:|:---:|---|:---:|:---:|:---:|:---:|:---:|
> > | method | 0.5s | 1s | 1.5s | 2s | mean |  | 0.5s | 1s | 1.5s | 2s | mean |
> > | Skeleton-graph [R4] | 91.4 | 153.9 | 222.9 | 313.7 | 162.7 |  | 98.8 | 107.2 | 112.2 | 116.8 | 106.1 |
> > | Ours | **58.0** | **103.2** | **154.9** | **221.7** | **108.4** |  | **50.8** | **67.5** | **75.5** | **86.9** | **61.4** |
> >
> > 5. Forecasting more than 5 seconds (question)
> >
> > __Response:__ We follow the setup in [6] to predict the future 60 frames (2 seconds) given the past 30 frames (1 second). To obtain motions in the further future, we iteratively feed the predicted future motion to the pretrained model. In the table below, we compare our results with the most competitive baseline DMGNN [18] on GTA-IM dataset. Our model still outperforms the baseline model in both path and pose error when predicting future motions up to 10 seconds. The large path and pose errors are expected because such long-term future is stochastic and should thus not be predicted with a deterministic model. As also mentioned by Reviewer EMAJ, human motion is multi-modal, especially for motion in the long-term future, e.g., more than 5 seconds. The most popular way of addressing this is to predict multiple possible future motions, i.e., stochastic human motion prediction, which will be part of our future work.
> >
> > |  |  |  | Path |  |  |  |  |  |  | Pose |  |  |  |
> > |:---:|:---:|:---:|:---:|:---:|:---:|:---:|---|:---:|:---:|:---:|:---:|:---:|:---:|
> > | method | 5s | 6s | 7s | 8s | 9s | 10s |  | 5s | 6s | 7s | 8s | 9s | 10s |
> > | DCGNN [18] | 1977.6 | 2334.2 | 2853.0 | 3245.3 | 3557.9 | 3938.4 |  | 171.0 | 180.3 | 197.4 | 203.2 | 214.3 | 217.9 |
> > | ours | **1970.7** | **2290.1** | **2751.0** | **3043.7** | **3439.7** | **3712.2** |  | **135.4** | **144.1** | **148.9** | **155.9** | **159.5** | **165.0** |
> >
> > 6. limitation about body surface contact (limitation)
> >
> > __Response:__ Thank you for the suggestion. Please see the revised manuscript for the updates. We will also add this to the limitation discussion in the final version.
> >
> > [R4] Mohamed, Abduallah, et al. "Skeleton-Graph: Long-Term 3D Motion Prediction From 2D Observations Using Deep Spatio-Temporal Graph CNNs." ICCV Workshop 2021.

---

### Official Review · Reviewer_EMAJ · 2022-07-12

**Rating:** 6
**Confidence:** 3
**Soundness:** 3 good
**Presentation:** 3 good
**Contribution:** 3 good

**Summary:**

This work proposes a two-stage human motion forecasting framework that explicitly models human-scene contact. It proposes to decouple the problem into two stages: past pose conditioned contact forecasting, and contact-conditioned pose forecasting. Specifically, it proposes to use a Discrete Cosine Transform (DCT) based network to predict contacts based on past contact, human pose, and scene point clouds. After the future contact is predicted, a series of networks are used to predict the human’s global translation, rotation, and body joint positions.

**Questions:**

It would be great if the authors could compare more closely with the motion generation literature (such as SAMP) and discuss differences.

**Limitations:**

The authors have discussed limitations adequately.

**Strengths And Weaknesses:**

## Strength

**Explicit Contact Modelling**

- The proposed two-stage pipeline is intuitive and performs well in the context of human motion prediction. Contact and physical constraints play an important role in governing human motion and based on contact human motion are a lot less ambiguous. The idea of explicitly predicting the future contact of humans in a known scene to guide motion prediction is interesting.

**Performance compared to State-of-the-art**

- The proposed method outperforms SOTA methods in the motion prediction task.

## Weakness

**Novelty in lieu of motion generation methods**

- Given the existence of methods such as SAMP [1], where human motion is generated based on path and scene context, the proposed framework has limited novelty. While the settings are slightly different (motion and interaction generation vs forecasting), the methodology is largely similar. The two-stage modeling has been largely explored (first generate goals or subgoals, then generate local motion), and this work mainly excels at better modality (explicit contact).

**Lack of generative modeling**

- While human motion is multi-modal, the lack of generative and stochastic modeling means that the estimated human motion could be memorizing past observed interactions (especially in PROX and GTA-IM datasets, where the motion are largely similar).

**Lacking Qualitative Results**

- Since motion is better seen in videos, it would be better if more qualitative samples are provided.

[1] Hassan, Mohamed et al. “Stochastic Scene-Aware Motion Prediction.” 2021 IEEE/CVF International Conference on Computer Vision (ICCV) (2021): 11354-11364.

---

> ### Author Response · Authors · 2022-08-02
> **Responses to Reviewer EMAJ**
>
> 1. Novelty in lieu of motion generation methods and comparison with SAMP in literature review.
>
> __Response:__ Although we both adopt multi-stage pipeline, the motivation and intention of each stage is different. SAMP [R1] first generates a goal location and orientation given a target object (goal generation). It then plans the path to the goal with searching techniques, such as A* (path planning). Finally, a motion net is used to generate a human pose at each frame. Our pipeline differs from that of SAMP [R1] in two ways. First, SAMP only considers interaction with a given object in the final frame, i.e, the goal, while our contact map prediction network does not rely on any object and aims to capture interactions with the entire scene in every frame. Second, SAMP's interaction representation is coarse, i.e., only a goal location and orientation. By contrast, our contact map models fine-grained relationships between every human joint and the scene. Such per-joint contact maps constrain both the global motion and the local human pose and can avoid issues like ''ghost motion''.
>
> Thank you for the suggestion. As shown in the revised manuscript, we will discuss SAMP [R1] in the parts of the text where we already review works on scene-aware human motion generation.
>
> 2. Lack of generative modeling.
>
> __Response:__ In this work, we tackle the problem of deterministic motion prediction. Therefore, given one history motion, we predict one future motion. Given a past motion, human movement in a short future is mostly deterministic because of physical constraints, e.g., Newton's laws. For example, a forward-walking person cannot suddenly turn backward. We acknowledge that human motion is multi-modal, especially for long-term future motions. This is addressed in the task of stochastic human motion prediction, which will be one of our future research directions.
>
> 3. Lacking Qualitative Results
>
> __Response:__ For qualitative motion comparisons, please see our supplemental video. We will include more comparisons in the final video.
>
> [R1] Hassan, Mohamed et al. “Stochastic Scene-Aware Motion Prediction.” 2021 IEEE/CVF International Conference on Computer Vision (ICCV) (2021): 11354-11364.

---

> > ### Comment · Reviewer_EMAJ · 2022-08-08
> > **Follow up to author response.**
> >
> > Thanks for the detailed response.
> >
> > Most of my concerns are addressed; the only remaining issue is the lack of qualitative results. I have checked the original included supplementary video and it only includes a handful of sequences (which overlaps with the plots included in the main paper). Some failure cases would also be useful to include in the video to better understand the results.

---

> > > ### Author Response · Authors · 2022-08-09
> > > **Additional results**
> > >
> > > Thanks for the suggestion.
> > >
> > > We have added additional results to the supplemental material. As also mentioned in the Checklist, we will release our source code upon the acceptance of this paper which also includes the code to visualize our results.

---

### Meta-Review · Area_Chair_JpYw · 2022-08-26

**Recommendation:** Accept
**Confidence:** Less certain

**Metareview:**

Three expert reviewers have recommended accepting the paper after the discussion period.  Reviewers like the overall idea and framework.  The AC agrees and recommends acceptance.  Please carefully revise the paper based on the reviews.

**Award:**

No

---

### Decision · Program_Chairs · 2022-09-14

Accept